# Zinc Plus Biopolymer Coating Slows Nitrogen Release, Decreases Ammonia Volatilization from Urea and Improves Sunflower Productivity

**DOI:** 10.3390/polym13183170

**Published:** 2021-09-18

**Authors:** Maqsood Sadiq, Usama Mazhar, Ghulam Abbas Shah, Zeshan Hassan, Zahid Iqbal, Imran Mahmood, Fahad Masoud Wattoo, Muhammad Bilal Khan Niazi, Atiku Bran, Kamusiime Arthur, Nadeem Ali, Muhammad Imtiaz Rashid

**Affiliations:** 1Department of Agronomy, Pir Mehr Ali Shah Arid Agriculture University, Rawalpindi 46000, Pakistan; mskhichi381@gmail.com (M.S.); shahga@uaar.edu.pk (G.A.S.); imran403@uaar.edu.pk (I.M.); atikubran@yahoo.com (A.B.); 1arthurk1@gmail.com (K.A.); 2College of Forestry, Sichuan Agriculture University, Chengdu 611120, China; usamamazhar333@gmail.com; 3Bahadur Sub Campus, College of Agriculture, Bahauddin Zakariya University, Layyah 31200, Pakistan; zeshan.hassan@bzu.edu.pk; 4Institute of Soil Science, Pir Mehr Ali Shah Arid Agriculture University, Rawalpindi 46000, Pakistan; zahidiqbal5245@gmail.com; 5Department of Plant Breeding & Genetics, PMAS-Arid Agriculture University, Rawalpindi 46000, Pakistan; fahad.pbg@uaar.edu.pk; 6Department of Chemical Engineering, School of Chemical and Materials Engineering, National University of Sciences and Technology, Islamabad 44000, Pakistan; m.b.k.niazi@scme.nust.edu.pk; 7Center of Excellence in Environmental Studies, King Abdulaziz University, Jeddah 21589, Saudi Arabia; nbahadar@kau.edu.sa

**Keywords:** biopolymer, coatings, microbial biomass, nitrogen losses, nutrient utilization efficiency, zinc coated urea

## Abstract

Currently, the global agriculture productivity is heavily relied on the use of chemical fertilizers. However, the low nutrient utilization efficiency (NUE) is the main obstacle for attaining higher crop productivity and reducing nutrients losses from these fertilizers to the environment. Coating fertilizer with micronutrients and biopolymer can offer an opportunity to overcome these fertilizers associated problems. Here, we coated urea with zinc sulphate (ZnS) and ZnS plus molasses (ZnSM) to control its N release, decrease the ammonia (NH_3_) volatilization and improve N utilization efficiency by sunflower. Morphological analysis confirmed a uniform coating layer formation of both formulations on urea granules. A slow release of N from ZnS and ZnSM was observed in water. After soil application, ZnSM decreased the NH_3_ emission by 38% compared to uncoated urea. Most of the soil parameters did not differ between ZnS and uncoated urea treatment. Microbial biomass N and Zn in ZnSM were 125 and 107% higher than uncoated urea, respectively. Soil mineral N in ZnSM was 21% higher than uncoated urea. Such controlled nutrient availability in the soil resulted in higher sunflower grain yield (53%), N (80%) and Zn (126%) uptakes from ZnSM than uncoated fertilizer. Hence, coating biopolymer with Zn on urea did not only increase the sunflower yield and N utilization efficiency but also meet the micronutrient Zn demand of sunflower. Therefore, coating urea with Zn plus biopolymer is recommended to fertilizer production companies for improving NUE, crop yield and reducing urea N losses to the environment in addition to fulfil crop micronutrient demand.

## 1. Introduction

The global population growth projection of 10 billion by 2050 will 70% increase the expected food demand [1]. A recent Food and Agriculture Organization report suggested that 2 billion people are suffering from food insecurity globally and COVID-19 pandemic further worsened the FAO goal of achieving Zero Hunger in 2030 [2]. Use of nitrogenous fertilizers in agriculture is reported to increase 30–50% of the global food production [3] and might able to meet the global food demand, if utilized smartly. Therefore, global requirement of these fertilizers increased by 6.67% (105 vs. 112 million tons) from 2016 to 2022, which would mainly be utilized in Asia [4]. Among these, urea is the dominant fertilizer type accounting 73.4% of the nitrogen (N) fertilizer demand [5]; however, this fertilizer is also subjected to high N losses through ammonia (NH_3_) volatilization. Global meta synthesis study showed that 64% of the applied N by urea in agriculture is lost to the environment through NH_3_ volatilization [6]. The future atmospheric warming and carbon dioxide (CO_2_) increment further increased this emission to 26.5% [7]. Leaching, surface runoff and nitrification from urea contributed 30% [8] and denitrification further add 30% of the N losses from this fertilizer to the environment [9]. The NH_3_ volatilization created acidification and eutrophication problems of oligotrophic habitats. In addition, N leaching and runoff from urea polluted ground and surface water bodies whereas nitrification-denitrification resulted in greenhouse gases emission and global warming [10,11]. Consequently, only 30–60% of the N applied through fertilizer is used by the plant [12], which is mainly due to poor N use efficiency (NUE) of the urea; hence, this resulted in low net crop productivity. Thus, there is a need to develop slow release urea fertilizers that have low N losses and high NUE [13] in order to meet the global future challenges of food insecurity and climate change.

Controlled release fertilizers provide nutrients to the plants in a controlled and delayed manner that if synchronize with plant needs, decrease nutrient emissions, leaching and runoff losses and also improve the crop NUE and yields [14]. For this purpose, numerous polymeric materials, natural coating agents, multifunctional super-absorbent materials, crop micronutrients such as sulfur and even nano-composites are being utilized for coating urea and other fertilizers [14,15]. Polymeric coatings are produced from the resins and thermoplastic whereas inorganic polymeric substances are derived from the graphene oxides gypsum, palygorskite sulphur and other materials [16]. These organic polymeric coated fertilizers showed significant reduction in NH_3_ volatilization and N leaching. Other polymeric materials such as polyethylene ((C_2_H_4_)_n_), polysulfone ((C_6_H_4_SO_2_)_n_), polyvinyl chloride (C_2_H_3_Cl)_n_, polystyrene (C_8_H_8_)_n_, polyacrylic acid latex (C_3_H_4_O_2_)_n_, chitosan, starch, polyvinyl alcohol ((C_2_H_4_O)_x_), mixture of gypsum (CaSO_4_), limestone (CaCO_3_), cement and zeolite (Na_2_Al_2_Si_2_O_8_) were also used for coating purpose in number of studies [17,18,19,20,21,22,23]. Nevertheless, most of these compounds did not decompose in the soil and remained there for long time, therefore becoming an additional source of organic and inorganic pollutants [24]. To overcome the soil pollution problem, several bio-based compounds including palm oil, animal manure, rice straw, lignin, starch [25], cellulose, honey wax, paraffin oil and gum arabica are used as encapsulation material for urea fertilizer [13,26,27,28,29,30,31]. However, most of these materials are expensive, non-durable and do not possess any nutrition value for plants.

Molasses is a bio-based compound obtained as a byproduct of the sugar production industry. It is dark brown, water soluble, thick liquid produced in the last stage of juice extraction from sugarcane [32]. Due to its soluble nature, this can be a suitable polymeric substance for coating materials. Recently, this material is used as binding agent during urea coating [12,14,32]. However, its suitability as urea coating agent has never been tested.

Superabsorbent polymers are the materials with hydrophilic cross-linking networks and possess a strong capability of huge amount of water absorption and retention. They do not only control urea release rate but also provide macronutrients such as sulfur or Zinc (Zn) to the plants if these nutrients are blended in the coating materials and thus improve the soil fertility [29]. Firstly, sulfur was used for coating urea to decrease its dissolution [33]. Sulfur is an essential nutrient required for crop growth and development at one hand and reduced the soil alkalinity effects on the other end [29]. However, its low wettability, adhesion, non-uniform coated layer on the granules and role in decreasing soil pH urged researchers to find efficient alternative solutions that can serve the same purpose but overcome the sulfur coating associated problems [14,29].

Zn is an essential nutrient and at the same time also acted as urease inhibitor for delaying the activity of urease enzymes [34]. However, this is a limiting nutrient for crop growth in tropical soils [35,36]. Accordingly, recent studies showed that >49% of the world arable soils in most of the tropical and temperate zone around the globe are Zn deficient [37,38]. This nutrient deficiency negatively affected plant growth, delayed their maturity and yield [37,39]. Hence, utilization of Zn as coating materials of urea has multifaceted benefits of improving the crop nutrition as well as slowing the N release rate from urea fertilizer [40]. Therefore, fertilizer coating technology should be designed in such a way that could not only synchronize the nutrients with plant demand but also fulfill the crop macro- and micronutrients requirement. Recently, Jiménez-Rosado*,* et al. [41] formed a superabsorbent bioplastic material using soy protein isolates to synchronize the Zn release with crop demand. Accordingly, they observed high release rate of Zn at initial stage since much of the zinc is required to horticultural crops at initial growth stage and much lesser quantity of this micronutrient is needed at later maturation stages. However, only few studies tested Zn suitability for urea coating [40,42,43]. To the best of our knowledge, no single study investigated the use of biopolymer plus Zn combination as coating material for urea fertilizer and release of Zn from the coated material in the soil. Moreover, no information is available on the development of bio-degradable polymer and micronutrient coated urea fertilizer and its micro and macro nutrients (Zn, N) release rate, nutrients crop uptake and NH_3_ volatilization at lab and field scale.

The main objective of the current study is to figure out, cheap, biodegradable and ecofriendly multi-purpose compounds as coating agents that have capability to slow down N release from urea fertilizer with provision of other micronutrients. The specific objectives of the study are (1) to characterize and evaluate the nutrients release rate of Zn plus biopolymer coated granular urea fertilizer (2) to study the effect of coated urea on ammonia emission, soil microbial and chemical properties and nutrients uptake by sunflower. We hypothesized that coating materials will fill the pores on urea granules by making a surface layer. This coating layer will control the urea release rate, decrease NH_3_ emission and synchronize the N release with crop N demand. Consequently, crop N uptake and N recovery will be higher in coated treatments than uncoated fertilizer.

## 2. Materials and Methods

Zinc sulphate hepta hydrate was purchased from Daejung Chemicals and Metals Company, Ltd., Shiheung, Korea. Molasses was obtained from Alumic Sugar Mills unit 2, Dera Ismail Khan, Pakistan. Commercial grade urea granular fertilizer was purchased from the Fauji Fertilizer, Bin Qasim Limited, Karachi, Pakistan.

### 2.1. Urea Coating through Fluidized Bed Coater

A 2.5 g zinc sulphate hepta hydrate (100 g of urea)^−1^ was added in 100 mL of de-ionized water. This was stirred on hot plat at a temperature of 80 °C for half an hour. In another mixture, same amount of zinc sulphate hepta hydrate and 5 g molasses (100 g of urea)^−1^ were added and mixed at 80 °C with constant stirring. These coating solutions were used in fluidized bed coater. Granular urea was coated by using the mini spray granulator YC-1000 (Shanghai Pilotech Instrument & Equipment Co., Ltd., Shanghai, China). Spray nozzle of the instrument was, located below the fluidized bed, containing the feed of urea grains. About 0.5 kg of product batch was run at a time. Urea grains were fluidized using hot fluidization air blown at 45 Hz through the heater at temperature of 80 °C. The built- in peristaltic pump was utilized for the movement of hot solution at 30 rpm. The hot coating solution was atomized using pressurized air through 0.2 MPa compressor before showering on fluidized urea grains in the bed. The coating procedure was only started once steady state temperature conditions were achieved and finished after 15 min of drying. Subsequently, the coated urea grains were removed from the bed and analyzed using different characterization techniques.

### 2.2. Characterization of Urea Granules before and after Coating

The morphology of the uncoated and coated urea granules was studied using a scanning electron microscopy (SEM, S-4700, Hitachi, Chiyoda City, Tokyo, Japan). Gold sputtering (250 angstroms) on urea grain was carried out by a JEOL JFC- 1500 anion sputtering machine (Jeol, Tokyo, Japan). The surface morphology of sample grain was determined by secondary electron detector using 20 Kv voltage at a magnification range of 25 to 500×.

Fourier Transform Infrared (FTIR) spectroscopy of the coated and uncoated urea granules was performed using FTIR Spectrometer (Perkin Elmer 710, Bridgeport Avenue, Shelton, CT, USA). The urea grains were broken into fine powder and examined in the wavelength range of 400 to 4000 cm^−1^. The resulted FTIR spectrum was used to evaluate the chemical structure and nature of bonding associated with different types of polymer material chains.

X-ray diffraction (XRD) characterizations of an uncoated urea grains and polymeric coated urea was carried out by using X-ray diffractometer from STOE, Darmstadt, Germany. This analysis was carried out to check the crystalline polymeric surface. The scan angle was ranged between 20° to 70°. Step size and step time were taken 0.4° and 1 s, respectively. The diffractometer had Ni-filtered Cu-Kα-1 rays with 0.15406 nm wavelength.

Crushing strength of the coated urea grain was analyzed for determining its capability to survive without breaking from production to sale and marketing phases. The excessive physical pressure during these phases might disintegrate urea grains and convert to fine powder. This fine form of urea is known as dust and it has no further use for agricultural activity. The crushing test of coated urea was done by using universal testing machine (AGX, Shimadzu, Kyoto, Japan). The coated urea grain was randomly selected from sample batch. Under testing, coated urea grains were put against force by means of metal plunger. The stress at which the urea grain disintegrated was noted as a measure of its strength [17].

### 2.3. Urea Release Kinetics

The release rate and efficiency of the urea coated with polymeric materials were evaluated by the Para-dimethyl amino benzaldehyde method. Initially, a calibration curve was drawn using analytical grade urea granules (99.9% purity). Standardized urea solutions were prepared to obtain the slope from the calibration curve. The absorbance of the standard solution was measured using UV-Visible spectrophotometry. To calculate the release rate of different formulations, the following procedure was adopted.

First of all, 10 g urea grains were taken in a 5-L glass beaker, diluted with de-ionized water. Then, 10 mL of sample aliquots were taken from the center of the beaker at time interval of 3, 6, 9, 12, 15, 30, 60 and 120 min. All samples were diluted with 50 mL distilled water and the absorbance was measured using UV-Visible spectrophotometer (GENSYS TM20, Thermo Fisher Scientific, Waltham, MA, USA). Before measurement, the solution in the beaker was mixed for 15 sec. After that, 10 mL sample was collected from the diluted sample and added in the 50 mL volumetric flask with 1 mL of HCL (1:1) and 5 mL of Para-dimethyl amino benzaldehyde solution. The de-ionized water was used to bring the final volume to 50 mL. Finally, the absorbance was noted using a wavelength of 418 nm to calculate the unknown concentration of the coated urea test batch using Equation (1). Afterward, concentration of coated and uncoated urea grains were used for calculating the urea release efficiency using Equation (2) [18].
(1)$$\mathrm{Urea}\;(\mathrm{ppm})=\frac{\left(\mathrm{Absorbance}\;-\;Y.\mathrm{Intercept}\right)}{\mathrm{Slope}\;\mathrm{from}\;\mathrm{calibration}\;\mathrm{curve}\;}$$
(2)$$\mathrm{Efficiency}\;(\%)=\frac{\mathrm{Cco}-\mathrm{Cun}}{\mathrm{Cun}\;}\times 100$$
where Cco and Cun are the concentration of urea (ppm) in the coated and uncoated urea sample, respectively.

### 2.4. Pot Experiment

A pot experiment was carried out with clay loamy soil at research facility of PMAS-Arid Agriculture University Rawalpindi, Pakistan (33.6492° N, 73.0815° E). The soil was taken from the top 30 cm fertile soil layer of the agricultural field. Afterward, the soil was sieved with 2 mm mesh to remove non-degradable material and plant debris and 13 kg soil was filled in each pot with a diameter of 26 cm. In total, 4 treatments (C = control, UCU = urea (uncoated), ZnS = Zinc sulphate hepta hydrate coated urea and ZnSM = Zinc sulphate hepta hydrate + molasses (5%) coated urea were applied in triplicates with complete randomized design. Both coated and uncoated urea were applied before sowing of seeds at the recommended doses of 100 kg N ha^−1^. To fulfill the potassium and phosphorus requirement, a basal dose of 70 kg P ha^−1^ (triple superphosphate) and 50 kg K ha^−1^ (potassium sulphate) was applied in fertilized pots. Thereafter, local sunflower cultivar (Parsan-3) was manually sown at the rate of 10 seeds pot^−1^. After germination, the plants were thinned to 5 plants/pot and the pots were placed at open space under natural environmental condition where soil moisture was maintained at 60%. Soil water content was continuously measured by cheap moisture meter (FY-901, Hangzhou FCJ I & E Co., Ltd., Hangzhou, China).

#### 2.4.1. Measurement of Ammonia Emission

Immediately after application of treatments, three passive samplers were installed above each pot for capturing ammonia (Figure 1).

The samplers were installed vertically at height of 20 cm above soil surface in the center of each pot for 72 h. Passive samplers were hanged in a wooden frame facing open side of each sampler downward. A distance of 15 m was kept between two adjacent pots to avoid mixing of NH_3_ between the experimental units. The NH_3_ was captured by steel grids coated with 60 µL of sulfuric acid (10% (*w*/*v*)). After 72 h, the samplers were taken out of wooden frame by immediately closing the opening with lids and placed in refrigerator at 4 °C [42]. For analysis, the steel grids were taken out from each sampler and rinsed with 5 mL deionized water. Subsequently, the NH_4_^+^-N content in the solution was analyzed as described in Shah, Shah, Rashid, Groot, Traore and Lantinga [42]). The average concentration of NH_3_ (ug m^−3^) was estimated according to the formula of Hofschreuder and Heeres [43]:(3)$${\mathrm{CNNH}}_{3}=\frac{{\mathrm{QNH}}_{4}{}^{+}\times \mathrm{Zlt}}{\begin{array}{c} \mathrm{Dco}\times At\times T \end{array}}\times \frac{17}{18\;}\;$$
(4)$$\mathrm{Dco}=\frac{\mathrm{Temp}\;1.75}{\mathrm{Pres}}\times \left(1.1265\times 10^{-9}\right)\;$$
where CNNH_3_ represents the concentration of NH_3_ (ug m^−3^), QNH_4_^+^ signifies the amount of NH_4_^+^-N measured in the deionized water (mg), Zlt is the tube length (0.041 m), Dco specifies the diffusion coefficient (0.0000247 m^2^/s/day) for above procedure. At represents the area of the inner tube (0.0000785 m^2^), T denotes the sampling time in s (seconds), Temp represents the air temperature in Kelvin (302.81) and Pres represents the air pressure (1.0023 bar).

#### 2.4.2. Soil Sampling and Analysis

Soil samples were taken from earthen pots at depth of 0 to 25 cm by using a manual augur. From each pot, soil was sampled from three random sites initially and at final harvest, the samples from each pot were combined to get a composite sample. These samples were analyzed for pH, dissolved organic carbon (DOC), Zn content, Nmin, plant available P & K, microbial biomass carbon (MBC), nitrogen (MBN) and Zn (MBZn). The pH was determined from soil and water (1:2.5) suspension using pH meter (inoLab pH Level 1, WTW GmbH & Co., KG, Weilheim in Oberbayern, Germany). After that, EC was determined from the same solution using an EC meter (DDS-12DW). Dissolved organic carbon (DOC) was measured by taking 5 g of soil and dissolved in 25 mL water. The suspension was placed in an incubator at 80 °C for 24 h and then DOC was determined using TOC analyzer [44]. Ammonium bicarbonate diethylenetriaminepentaacetic acid (AB-DTPA) procedure was used to determine soil mineral N (nitrate-N and ammonium-N). Soil available P and K were calculated according to method described in Houba*,* et al. [45]. The soil used in the study had a pH of 8.3 ± 0.02, EC (2.8 ± 0.17 dSm^−1^), mineral N (3.6 ± 0.64 mg kg^−1^), plant available P (2.5 ± 0.31 mg kg^−1^), K (197 ± 4.0 mg kg^−1^), Zn (1.54 ± 0.05 mg kg^−1^) and DOC (3.84 ± 0.34%).

#### 2.4.3. Microbial Biomass Carbon, Nitrogen and Zinc

Fumigation extraction method was used for analysis of MBC and MBN in the soil. For this purpose, 10 g of soil was weighed and equally divided into two parts. One part of soil was fumigated in ethanol free chloroform for 24 h whereas the other part kept as non-fumigated. After fumigation, soil sample was placed in hot water bath at 80 °C for two hours. Subsequently, 25 mL of 0.5 M K_2_SO_4_ solution was added in both fumigated and non-fumigated soils and shaken for half an hour. Afterwards, soil solution was filtered through Whatmann no. 42 filter paper. Total organic C and N in filtrates were analyzed using TOC analyzer and Kjeldahl digestion method. Soil MBC and MBN were calculated by using following equation:(5)$$\mathrm{MBC}\;\mathrm{or}\;\mathrm{MBN}=\frac{{\mathrm{TOC}}_{f}{\;\mathrm{or}\;\mathrm{TN}}_{f}-{\;\mathrm{TOC}}_{\mathrm{nf}}{\;\mathrm{or}\;\mathrm{TN}}_{\mathrm{nf}}}{\mathrm{kETOC}\;\mathrm{or}\;\mathrm{kETN}}$$
where TN_f_ and TN_nf_ is total nitrogen in fumigated and non-fumigated soil samples, respectively, while TOC_f_ and TOC_nf_ is total carbon in fumigated and non-fumigated soil samples, respectively. kETOC (0.45) and kETN (0.54) were used as coefficient for calculation of MBC [46] and MBN, respectively [47,48].

The above-mentioned method was used for measurement of microbial biomass Zn. However, after treating the fumigated soil in hot water bath, 25 mL of 1 M NH_4_NO_3_ was added in both fumigated and non-fumigated soil samples as a substitute of K_2_SO_4_. These samples were shaken for 30 min on a mechanical shaker and then filtrated through Whatmann filter paper no. 42. The filtrate was acidified with suprapur HNO_3_. The labile Zn concentration in soil extract was analyzed by using atomic absorption spectrophotometer (Hitachi Polarized Zeeman, ZA3000 Series, Chiyoda City, Tokyo, Japan). Microbial biomass Zn (MBZn) was calculated by using following equation:(6)$$\mathrm{MBZn}={\mathrm{Zn}}_{f}-{\mathrm{Zn}}_{\mathrm{nf}}$$
where Zn_f_ and Zn_nf_ is the amount of Zn in fumigated and non-fumigated soil samples, respectively.

#### 2.4.4. Plant Growth Attributes

Sunflower was harvested at physiological maturity stage. The plant growth attributes i.e., plants height (cm), number of leaves per plant, stem girth (cm) and leaf area (cm^2^) were determined weekly until physiological maturity stage reached. Content of the leaf chlorophyll was calculated before harvest using a SPAD chlorophyll meter (Konica Minolta Sensing Europe B.V., MR Nieuwegein, The Netherlands). At the end of experiment, the plants were harvested and placed in an oven at 70 °C for 48 h. Subsequently, plant dry matter yield, head diameter (cm), head dry weight (g plant^−1^), no. of seed head^−1^ and seed yield (kg ha^−1^) were determined. All dried plant samples were mixed to make a composite sample and N content were determined using Kjeldahl digestion method. Plant N and Zn uptakes were calculated by using following equation.
(7)$$\mathrm{Apparent}\;N\;\mathrm{and}\;\mathrm{Zn}\;\mathrm{Recovery}\;(\%)=\frac{(N_{\mathrm{mo}}{\;\mathrm{or}\;\mathrm{Zn}}_{\mathrm{mo}}\times {\;\mathrm{DM}}_{f})-\left(N_{0}\;{\mathrm{or}\;\mathrm{Zn}}_{0}\times {\;\mathrm{DM}}_{0}\right)}{{\mathrm{TN}}_{\mathrm{po}}{\;\mathrm{TZn}}_{\mathrm{po}}}$$ where N_mo_ and Zn_mo_ is the amount of nitrogen and zinc applied in fertilizer pots (kg N, kg Zn). DM_ms_ represents dry matter yield of respective fertilizer pots (kg ha^−1^). N_0_ and Zn_0_ shows nitrogen and zinc in control treatment of soybean. DM_mo_ represents a dry matter yield of soybean in control (kg ha^−1^). TN_po_ and TZn_po_ is the amount of nitrogen and zinc applied in pots (kg ha^−1^).

### 2.5. Statistical Analysis

The treatments effects were analyzed by univariate analysis using IBM SPSS Statistics version 26 (New York, NY, USA). The main effects of treatments were analyzed statistically through analysis of variance (ANOVA) technique. The means of all treatments were compared at 5% probability level. When treatment effects were significant, then multiple comparisons among treatments were analyzed by Tukey’s HSD test.

## 3. Results

### 3.1. Surface Morphology and Characterization of Coated and Uncoated Urea Fertilizer

The morphological analysis indicated a significant difference in the coated and uncoated urea granules (Figure 2a–c). Both coated treatments (ZnS & ZnSM) did not have any significant gaps or pinholes on the surface layer and showed homogenous distribution of zinc sulphate coating on the surface. Zinc sulphate plus molasses (ZnSM) had more uniform and smoother surface than only zinc sulphate coated urea. Molasses was able to improve the overall uniformity by forming a compact layer of zinc sulphate coating on urea granules (Figure 2c).

The Fourier transform infrared (FTIR) spectra of all coated urea treatments were similar to the uncoated one with slight differences in the intensity of corresponding spectral peaks (Figure 2d). These spectra indicated the presence of transmittance peaks at 3447 cm^−1^ and 3343 cm^−1^ representing the asymmetric and symmetric free stretching vibrations of –O-H groups. Similarly, the peak appeared at 2810 cm^−1^ showed a stretching vibration of N-CH_3_ aromatic group. In addition to this, the transmittance peak at 1688 cm^−1^ and the peak at 1613 cm^−1^ showed the presence of C=C stretching and NH bending, whereas the transmittance at 1425 cm^−1^ and 1154 cm^−1^ signifies O-H bending and C-O vibrational stretching. The C=C and C-O indicated that organic compound such as molasses was present in ZnSM treatment. However, there was not much difference in FTIR peaks of the coated and uncoated urea treatments. Hence, these characteristics confirmed that the coated treatments spectra were similar to the uncoated urea which suggests that binding of coating material and urea were taken place through physical bonds, i.e., Van der Waals and hydrogen bonding forces whereas no chemical reaction occurred to form any undesirable compounds during the mixing and coating process. The diffractogram of uncoated and coated urea treatments are shown in Figure 2e. Similar to FTIR analysis, the coated urea diffractograms showed no shift in peaks location or strength compared to the diffractogram of uncoated urea. However, a slight difference in the peaks intensities was observed among coated and uncoated treatments. This suggests that the coating materials were attached with urea granules through physical bonds; however, the overall structure of the coated urea granules was not modified substantially. In all treatments, the diffractogram designated 2θ Braggs’ reflection at 22°, 24.5°, 29.5° and 35°. All coated samples had sharp peaks than uncoated urea showing high crystallinity and suggesting that a clear coating has been formed on the surface of urea granules.

### 3.2. Effect of Coating on Urea Release and Efficiency

Urea release rate and its efficiency are presented in Figure 3. In the UCU treatment, urea was completely released in water after 15 min and 12 s of immersion.

The urea from ZnS treatment was released completely after 60 min and 31 s. On the other hand, the complete release of urea from ZnSM treatment was observed in 140 min and 15 s. Therefore, the highest urea release efficiency was observed in ZnSM (44%) treatment and the lower in ZnS treatment (25%).

### 3.3. Ammonia Emission

Ammonia emission was significantly affected by the treatments (Figure 4A). Uncoated urea fertilizer increased NH_3_ emission by 52% (108 vs. 71 µg m^−3^) compared to control. Multiple comparison indicated that the ZnS coating did not significantly decease this parameter from uncoated urea. However, ZnSM coatings decreased NH_3_ emission by 38% (78 vs. 108 µg m^−3^) than uncoated urea. Interestingly, there was no difference in NH_3_ emission between control and ZnSM treatment indicating that this coating combination was able to reduce most of the losses occurred through NH_3_ emission from the urea fertilizer (Figure 4A).

### 3.4. Soil Parameters

Soil pH was significantly affected by urea fertilizer treatments (Figure 4B). All fertilizer treatments significantly decreased soil pH. Uncoated urea fertilizer decreased this parameter by 5% (8.2 vs. 8.6) compared to control and this decrement was 8% in ZnS and 7.6% in ZnSM treatments. Moreover, ZnS significantly decreased soil pH compared to uncoated urea but this parameter was not different between UCU and ZnSM treatments (Figure 2B). All urea fertilizer treatments significantly increased the dissolved organic carbon (DOC) in the soil compared to control treatment (Figure 5A). Moreover, the multiple comparison showed no significant differences among uncoated and coated urea treatments. The soil mineral N was higher in all urea treatments compared to control (Figure 5B). This parameter was 21% higher in ZnSM treatment than uncoated urea but did not differ between ZnS and uncoated as well as ZnS and ZnSM treatments (Figure 5B). The plant available P (PAP) was not affected by urea and ZnS coated urea treatments. Similarly, PAP was not different between ZnSM and ZnS treatments. This parameter was 41% higher in ZnSM treatment compared to the control. Similarly, PAP did not differ significantly among the ZnS, UCU and C treatments. Soil Zn was not significantly affected by the treatments (*p* > 0.05: Figure 5D). Although this parameter was tended to be higher in ZnS and ZnSM treatments compared to uncoated urea, but it did not differ statistically in these treatments.

Microbial biomass C and N was significantly affected by the treatments (Figure 6A). Interestingly, MBC was not different among control, uncoated and ZnS coated urea treatments. However, MBC was 4× higher in ZnSM treatments compared to uncoated urea. On the other hand, MBN was significantly higher in ZnS and tended to be high in ZnSM treatments compared to uncoated urea and control. However, there was no difference in this parameter between coated urea treatments (Figure 6A). In non-fumigated soils, the Zn content did not differ among treatments. However, in fumigated soils, Zn was >2× higher in ZnSM than uncoated urea treatment but this parameter in ZnS treatment was not different from uncoated urea (Figure 6B).

### 3.5. Plant Growth and Yield Attributes

The plant growth and yield attributes were also significantly affected by the treatments. Uncoated urea fertilizer improved leaves chlorophyll content. Harvest index and stem diameter were differed significantly among treatments. The leaf chlorophyll content was 21% higher in uncoated urea treatment than control (Table 1). Multiple comparison indicated that chlorophyll content in leaf was not differed significantly among coated and uncoated urea treatments, although this parameter was significantly higher in all fertilizer amended treatments than control. Similarly, plant height, fresh head weight and head diameter was significantly higher in coated and uncoated urea treatments than control. However, there is no difference in these parameters among uncoated and coated urea treatments. Leaf area index, number of leaves per plant and 100 seed weight were not differed between uncoated urea and control treatments but these parameters were significantly higher in ZnS and ZnSM treatments than both control and uncoated urea treatments. However, these parameters were not different between ZnS and ZnSM treatments. Seed and root dry matter yield did not differ between control and uncoated urea treatment. This parameter was highest in the ZnSM treatment but did not differ between ZnS and uncoated urea treatments (Table 1). Grain yield followed similar trend, this was the highest in ZnSM, followed by ZnS and uncoated urea treatment. This parameter was 53 and 36% higher in ZnSM and ZnS treatments compared to uncoated urea, respectively, showing that ZnSM treatment outperformed in grain yield than ZnS, uncoated urea and control treatments (Table 1).

Sunflower N and P uptakes were significantly affected by treatments (Figure 7A). There was no significant difference in these parameters among control, uncoated urea and ZnS coated urea treatments as revealed by multiple comparison test. However, both aforesaid nutrients were 80 and 78% higher in ZnSM than uncoated urea treatment, respectively. Plant Zn uptake was also significantly affected by the treatments. This parameter was 126% higher in ZnSM and 118% in ZnS treatments than uncoated urea. Surprisingly, Zn uptake was 109% higher in uncoated urea than control treatment indicating urea application also increased micronutrients, i.e., Zn uptake (Figure 7B).

## 4. Discussion

In line with our first hypothesis, we observed significant differences in the surface morphology of coated and uncoated urea granules (Figure 2a–c). Both ZnS and ZnSM treatments showed no gaps or pinholes on the granules surface and there was a homogenous distribution of ZnS coating (Figure 2). Interestingly, polymer (molasses) with ZnS had much smoother and uniform surface than ZnS coated urea which had still some fractures left on the surface (Figure 2b,c). According to Beig, Niazi, Jahan, Kakar, Shah, Shahid, Zia, Haq and Rashid [13]) coating with molasses in addition to other polymers increased the surface uniformity of the urea granules. Similarly, Irfan, Khan Niazi, Hussain, Farooq and Zia [40]) also observed that molasses and paraffin oil increased the uniformity of Zn coated urea granules. In line with these studies, we observed that molasses plus Zn formed a compact layer on the urea granules surface and, hence, improved the granules uniformity (Figure 2c). The success of the coating materials was also confirmed by FTIR spectra analysis that indicated that the spectra of all coated treatments were similar to the uncoated urea with slight differences in the intensity of the spectral peaks among coated and uncoated treatments (Figure 2d). The transmittance peaks at 3447 cm^−1^ and 3343 cm^−1^ indicated –O-H and amine (-NH_2_) groups showing the presence of water and urea [13,40]. Similarly, the peak appeared at 2010 cm^−1^ and 2200 cm^−1^ showed the presence of Alkyne (C≡C) and nitrile (C≡N) groups [49]. The presence of the aforementioned stretching vibrations are also the indications of better N release from the urea fertilizer [12]. Similar to FTIR analysis, X-ray diffractogram of the coated and uncoated urea granules showed no much difference or shift in peaks in the location or strength (Figure 2e). A slight difference in the peak intensities suggested that the coating materials were attached to the urea granules through physical bonds and, therefore, the overall structure of the granules was not modified substantially. The presence of sharp peaks on both coated granules than uncoated treatment displaying high crystallinity suggesting that a clear coating has been formed on the surface of urea granules [12,14,41].

According to our second hypothesis, we observed that both coatings decreased the NH_3_ emission from urea after its soil application, although it was not statistically significant in case of ZnS treatment (Figure 4A). Our findings are in line with Klimczyk*,* et al. [50]) who observed that controlled release urea fertilizer decreased NH_3_ emission by 30–70% in laboratory and field based studies. Similarly, Zhang*,* et al. [51]) revealed in a meta-analysis study that controlled release fertilizer decreased the NH_3_ emission by nearly half of the uncoated urea. They explained that slower and gradual release of N from controlled release urea fertilizer could be the most probable cause of lower NH_3_ volatilization than uncoated urea. Huang*,* et al. [52]) observed that increase in soil pH caused an increment in the liquid phase of NH_3_ rates which led to more volatilization of NH_3_ gas. In our study, the coated urea treatments decreased soil pH by 3% compared to uncoated urea (Figure 4B). This decrement in soil pH could be related to decrease in NH_3_ volatilization in our study as was observed by Huang, Lv, Bloszies, Shi, Pan and Zeng [52]). Secondly, the slow release of N from coated urea (Figure 3A) could result in the lower concentration of NH_4_^+^ ions in the soil solution and may decrease the partial pressure of NH_3_ in gaseous phase after short time of N application [53]. Therefore, the ZnSM coated urea could maintain the low N concentration in the soil solution during initial short duration due to decrease in soil pH and controlling the release of N than uncoated urea treatment in our study. Hence, the NH_3_ volatilization was lower in this treatment in our study than uncoated urea fertilizer.

Our expectation that coated urea treatments control N release from the fertilizer was in line with our N release kinetics result, where we observed slower N release from coated than uncoated urea after its water dissolution (Figure 3A). Among the coated treatments the higher release of N in water from ZnS than ZnSM treatment might be attributed to Zn water solubility [54]. On the other hand, the adhesion property of molasses as biopolymer created a uniform coating on the urea granules with the Zn and, therefore, such coating decreased the granule water solubility [13,40]. Moreover, Zn is considered as urease inhibitor; therefore, it can decrease the NH_4_^+^-N release from the urea [55]. Therefore, in our study the biopolymer plus Zn coating could able to decrease urea hydrolysis and slowed down the NH_4_^+^-N release from urea as was observed in case of Noor Affendi, Yusop and Othman [55]). Consequently, we observed lower N release rate from urea in ZnSM treatment than ZnS or uncoated urea (Figure 3A).

Our final hypothesis that coated urea granules will slowly increase N availability in the soil that will synchronize with sunflower N demand; therefore, we will observe higher sunflower yield, N and Zn uptakes than uncoated urea. We observed high mineral N content in the soil even at the end of the pot experiment (Figure 5B) indicating that ample concentration of mineral N was available for crop uptake during the crop growth stages. Moreover, the slower release of N in water from coated urea treatments than uncoated fertilizer (Figure 3A) confirmed that N in the coated urea treatments might release slowly in the soil and synchronize with the sunflower N demand. Interestingly, microbial biomass C and N was also higher in ZnS and ZnSM coated treatments than uncoated urea suggesting that high availability of mineral N in the soil favors microbial growth (Figure 5B). The soil DOC in the ZnS and ZnSM treatments was also tended to be higher than uncoated urea (Figure 5A). Montaño, et al. [56]) observed an increase in microbial biomass and activity with increasing dissolved organic N and C in the soil. In line with these findings Li*,* et al. [57]) observed that urea fertilization changed the microbial community which is closely linked to the soil C and nutrient cycling. Consequently, the higher mineral N and DOC in the soil of coated urea treatments (Figure 5A,B) may increase the microbial biomass C and N (Figure 6A) and, thus, influenced the nitrogen cycling [57] in our study.

As expected, slow release of N from the coated fertilizer resulted in higher dry matter and grain yield, N and Zn uptakes than uncoated urea fertilizer (Table 1, Figure 7). The coating material Zn has a capability to inhibit urease enzyme and, therefore, it can control the NH_4_^+^-N release from the urea [55]. Moreover, the addition of molasses, a strong adhesion agent made a uniform coating of ZnS plus molasses on the urea granule surface and, thus, further reduced the N release rate [13,40]. Such control release of N might have synchronized with the sunflower N demand when required at different stage of its growth, therefore, this resulted in higher growth and yield attributes of sunflower (Table 1) and N and Zn uptakes from the coated urea than uncoated fertilizer (Figure 7). Interestingly, there was no difference in sunflower Zn uptake between ZnS and ZnSM treatments but Zn uptake in these treatments was higher than uncoated urea and control despite of no difference in soil Zn among all treatments at the end of experiment (Figure 5D). Moreover, microbial biomass Zn was higher in ZnSM than all other treatments (Figure 5B) showing that molasses played an important role in controlling the release of both Zn and N from ZnSM treatment. Therefore, a controlled release of Zn was available for plant when required for its physiological growth parameters [40] and at the same was sufficient enough to support microbial biomass in the soil that might be the reason we observed higher microbial biomass Zn and sunflower Zn uptake in this treatment than uncoated urea (Figure 6B and Figure 7B). In line with our study Dimkpa*,* et al. [58]) also observed that Zn fertilization increased Zn uptake in sorghum without influencing other nutrients, such as N, P and K in the soil. This suggested that N fertilization did not have any synergistic or antagonistic influence on crop Zn uptake but it is the Zn applied as fertilizer that would resulted in higher Zn uptake by plants. That might be the reason, we observed higher Zn uptake in both ZnS and ZnSM treatments than uncoated urea despite of higher soil N availability in ZnSM than uncoated and no difference of N in ZnS and uncoated urea treatment (Figure 5B).

## 5. Conclusions

This is the pioneering study presenting a detailed analysis of Zn and biopolymer coated urea fertilizer on controlled N release rate, NH_3_ loss, soil N availability and sunflower N and Zn uptake in a laboratory scale and pot experiments. Despite of Zn urease inhibition activity, Zn sulphate coated urea did not able to reduce NH_3_ losses, control much the N release rate, soil N availability, microbial biomass C, N and Zn. However, molasses (biopolymer) and Zn coated urea controlled the N and Zn release rate, reduced NH_3_ losses and synchronized these controlled release nutrients with sunflower demand; therefore, we observed higher N and Zn availability in the soil, microbial biomass C, N and Zn as well as sunflower grain and total biomass yield, N and Zn uptake in this treatment. This could be ascribed to the great adhesion ability of molasses that in combination with Zn might decrease urease activity, urea hydrolysis and ultimately controlled both nutrients Zn and N release rates. Hence, it is recommended to the farmers and fertilizer companies to coat urea with biopolymer and Zn in order to increase the fertilizer N utilization by the crops and reduce N losses and meet the crop micronutrient i.e., Zn demand.

## Figures and Tables

**Figure 1 polymers-13-03170-f001:**
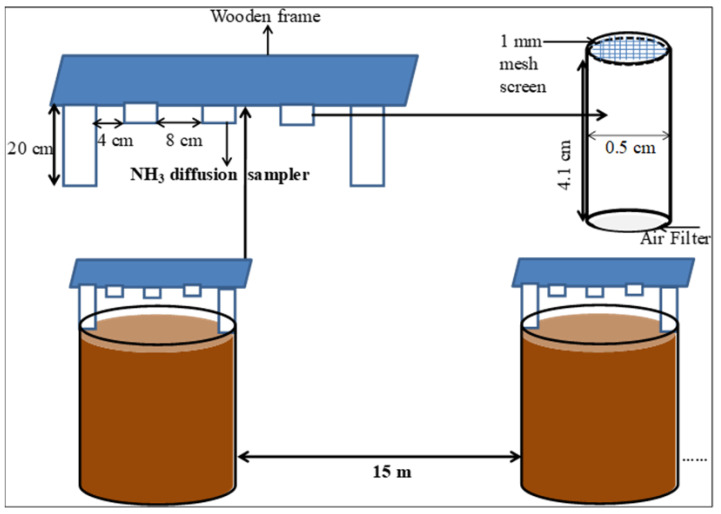
Schematic diagram of ammonia emission (NH_3_) measurement from the pots.

**Figure 2 polymers-13-03170-f002:**
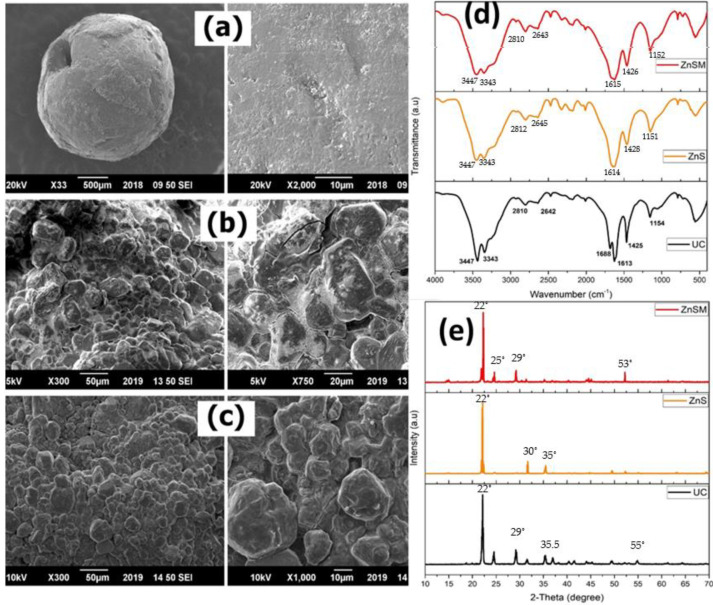
SEM micrographs of uncoated urea fertilizer (**a**), coated with zinc sulphate (**b**) and zinc sulphate + molasses (**c**). Fourier transform infrared (FTIR) spectra (**d**) and X-ray diffractogram (**e**) of coated and uncoated urea fertilizers.

**Figure 3 polymers-13-03170-f003:**
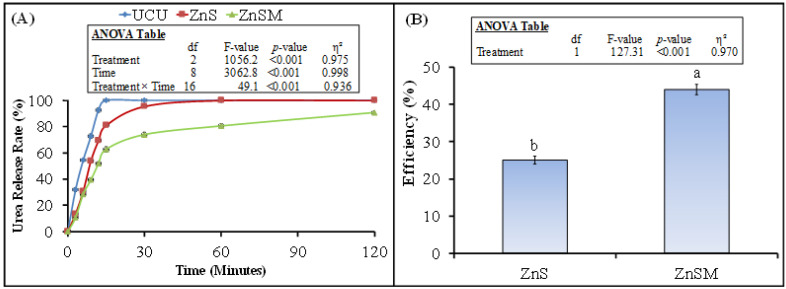
Urea release rate (%) (**A**) and its efficiency (**B**) from uncoated and zinc sulphate (ZnS) and zinc sulphate plus molasses (ZnSM) coated urea. Different small letters in Figure 3B show significant difference between treatments at 5% probability level. Inset represents ANOVA table. The comparison among treatments was analyzed by T-test.

**Figure 4 polymers-13-03170-f004:**
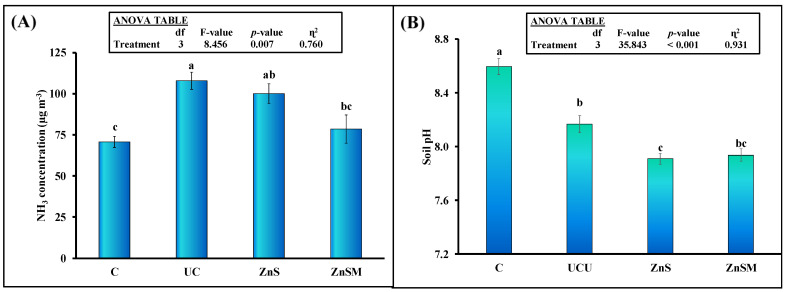
Mean (*n* = 3) NH_3_ concentration (µg m^−3^) (**A**) and soil pH (**B**) after application of uncoated urea (UCU) and urea granules coated with zinc sulphate (ZnS) and zinc sulphate plus molasses (ZnSM). C stands for untreated control treatment. Error bars showed standard errors (±1 SE) of the mean. Different small and capital letters show significant difference at 5% probability level. Inset represents ANOVA Table. Multiple comparisons among treatments were analyzed by Tukey’s HSD test.

**Figure 5 polymers-13-03170-f005:**
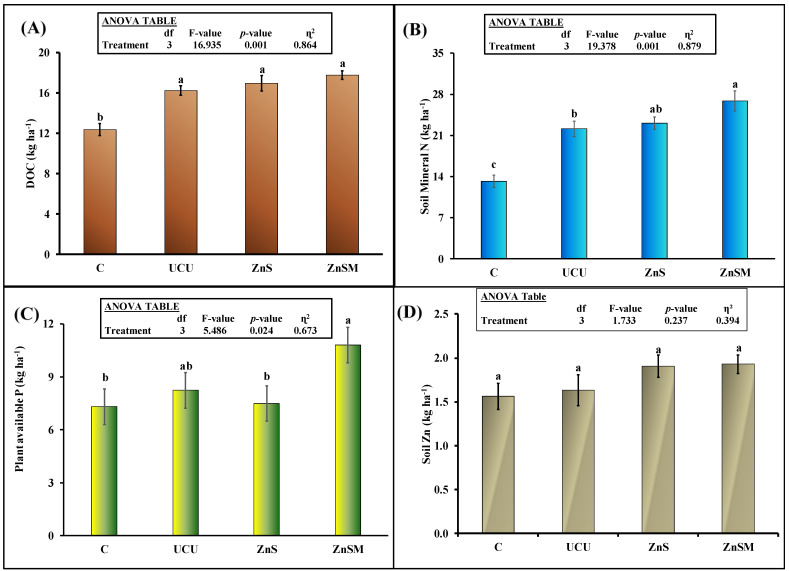
Mean (*n* = 3) soil organic carbon (DOC) (**A**), mineral nitrogen (**B**), plant available phosphorous (**C**) and zinc (**D**) as affected by application of uncoated (UCU) and coated urea with zinc sulphate (ZnS) and zinc sulphate plus molasses (ZnSM). C represents untreated control. Error bars showed standard errors of the means. Different small letters represented significance differences among treatments at 5% probability level. Inset shows ANOVA Table. Multiple comparisons among treatments were analyzed by Tukey’s HSD test.

**Figure 6 polymers-13-03170-f006:**
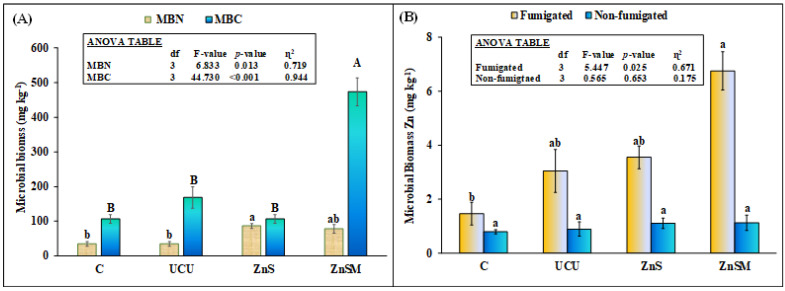
Mean (*n* = 3) microbial biomass carbon (MBC), nitrogen (MBN) (**A**) and zinc (MBZn) (**B**) in non-fumigated and fumigated soil after application of uncoated (UCU) and coated urea with zinc sulphate (ZnS) and zinc sulphate plus molasses (ZnSM). C represents untreated control. Error bars showed standard errors of the mean. Different small and capital letters show significant differences among treatments at 5% probability level. Inset in figure represents ANOVA Table. Multiple comparisons among treatments were analyzed by Tukey’s HSD test.

**Figure 7 polymers-13-03170-f007:**
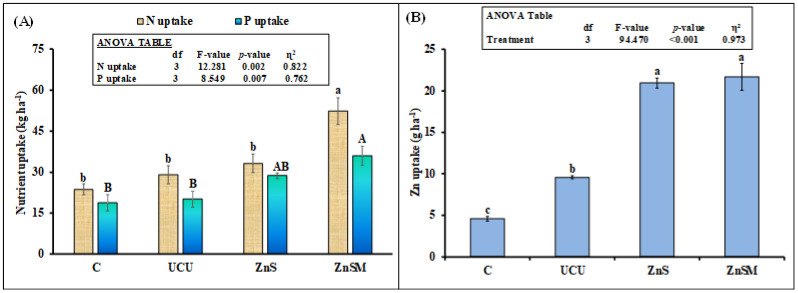
Mean (*n* = 3) crop N, P (**A**) and Zn uptakes (**B**) from uncoated (UCU) and coated urea with zinc sulphate (ZnS) and zinc sulphate plus molasses (ZnSM). C stands for untreated control treatment. Error bars showed standard errors (±1 SE) of the mean. Different small and capital letter shows significance at 5% probability level. Inset in figure represents ANOVA Table. Multiple comparisons among treatments were analyzed by Tukey’s HSD test.

**Table 1 polymers-13-03170-t001:** Mean (*n* = 3, ±1 SE) growth and yield parameters of sunflower crop as influenced by uncoated urea (UCU), coated urea with zinc sulfate (ZnS) and zinc sulfate plus molasses (ZnSM). C stands for untreated control. Different small letters within a column indicates the significant difference among treatments at 5% probability level after Tukey HSD test.

Treatments	Chl	PH	SD	NOL	LAI	FHW	HD	NSH	100SW	GY	SDMY	RDMY	HI
	(SPAD)	(cm)			(cm^2^ cm^−2^)	(g plant^−1^)	(cm)		(kg ha^−1^)				(g)
C	32 ^b^ ± 1.0	136 ^b^ ± 3.3	2 ^a^ ± 0.07	17 ^b^ ± 0.2	1 ^b^ ± 0.1	32 ^b^ ± 1.7	17 ^b^ ± 1.0	128 ^c^ ± 5	4 ^b^ ± 0.1	2044 ^a^ ± 58	1526 ^c^ ± 47	609 ^b^ ± 99	57 ^a^ ± 1.8
UCU	39 ^a^ ± 1.3	158 ^a^ ± 3.7	2 ^a^ ± 0.2	17 ^b^ ± 0.4	1 ^b^ ± 0.1	43 ^a^ ± 1	23 ^a^ ± 1.1	159 ^b^ ± 6	4 ^b^ ± 0.1	2649 ^b^ ± 128	1947 ^bc^ ± 129	647 ^b^ ± 50	58 ^a^ ± 3.8
ZnS	38 ^a^ ± 1.1	168.3 ^a^ ± 2.7	2 ^a^ ± 0.06	19 ^a^ ± 0.3	2 ^a^ ± 0.2	44 ^a^ ± 1.1	23 ^a^ ± 0.9	189 ^a^ ± 5	5 ^a^ ± 0.1	3594 ^c^ ± 68	2060 ^ab^ ± 120	854 ^ab^ ± 41	60 ^a^ ± 1.4
ZnSM	39 ^a^ ± 1.3	164 ^a^ ± 5.45	2 ^a^ ± 0.05	19 ^a^ ± 0.7	2 ^a^ ± 0.1	44 ^a^ ± 1.3	23 ^a^ ± 1.0	202 ^a^ ± 5	6 ^a^ ± 0.2	4057 ^d^ ± 46	2167 ^a^ ± 12	1062 ^a^ ± 45	60 ^a^ ± 3.4

Chl = chlorophyll, PH = Plant height, SD = Stem diameter, NoL = Number of leaf per pot, LAI = Leaf area index, FHW = Fresh head weight, HD = Head diameter, NSH = Number of seed per head, SW = seeds weigh, GY = Grain yield, SDMY = Shoot dry matter yield, RDMY = root dry matter yield, HI = Harvest index.

## Data Availability

All data is presented in the manuscript.

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
