# Peer review of "Zinc Plus Biopolymer Coating Slows Nitrogen Release, Decreases Ammonia Volatilization from Urea and Improves Sunflower Productivity"

_polymers, 2021, doi:10.3390/polym13183170_

Round 1

Reviewer 1 Report

The present study is examining the Zn plus biopolymer coating on N release on sunflower crop

The present study is of interest and innovative, the introduction and M&M are complete. There are some formatting issues and authors can address those in their revised version. The results are appropriately presented, the figure is suggested to be more simply in colored (no need for multiple colors, as it is confusing for differences among the treatments). The Discussion section is well described, with critical views.

To my opinion, the present work needs minor revision

This can be done also during authors proofreading, if the manuscript is accepted for publication

Some suggestions/comments

  • Check ‘’Zincated’’
  • Check chemical format. Same at L180
  • Please specify if possible the macronutrients referred to in that study
  • You may specify the crops in general that you are referring to. For example, in Aromatic plants, Zn has been proven to be of importance even at a later stage, especially on the essential oil composition/quality.
  • Check units. Should be ‘’mL’’ and not ‘’ml’’. check that throughout the text
  • Provide city and country for the equipment
  • Provide full name for all the abbreviations first time mentioned in the text.

Author Response

Response to Reviewer#1 Report

Manuscript ID: polymers-1365278

Title: Zinc plus biopolymer coating slows N release, decreases ammonia volatilization from urea and improves sunflower productivity

Dear Editor,

On behalf of the co-authors, I appreciate and thankful for the useful comments and suggestions raised by the respected reviewers #1 and #2 on the previous version of the manuscript. Hereby, I am resubmitting the manuscript after fully taking into account the comments and questions raised and the corrections proposed by the reviewers. The corrections are presented in track changes option in the manuscript. Authors are very thankful to the reviewers and editor for their positive comments. The point-by-point answer to the comments and questions of the reviewers could be found below:

Reviewer #1

The present study is examining the Zn plus biopolymer coating on N release on sunflower crop

The present study is of interest and innovative, the introduction and M&M are complete. There are some formatting issues and authors can address those in their revised version. The results are appropriately presented, the figure is suggested to be more simply in colored (no need for multiple colors, as it is confusing for differences among the treatments). The Discussion section is well described, with critical views.

Response: We are very thankful to the respected reviewer for the positive remarks about our study and the manuscript. As suggested, we remove the multi-color in figures where necessary.

To my opinion, the present work needs minor revision. This can be done also during authors proofreading, if the manuscript is accepted for publication

Response: Thank you very much for your insightful remarks about the manuscript.

 Some suggestions/comments

  • Check ‘’Zincated’’

Response: We revised to zinc coated urea.

  • Check chemical format. Same at L180

Response: The chemical format is revised now to make it similar. Please consult lines 181 and 193.

  • Please specify if possible the macronutrients referred to in that study

Response: There was a mistake in the sentence which is now revised as “They do not only control urea release rate but also provide macronutrients such as sulfur or Zinc (Zn) to the plants if these nutrients are blended in the coating materials and thus improve soil fertility”. Please consult lines 91-93.

  • You may specify the crops in general that you are referring to. For example, in Aromatic plants, Zn has been proven to be of importance even at a later stage, especially on the essential oil composition/quality.

Response: We added horticultural crops as was described in this cited publication. Please consults line 112.

  • Check units. Should be ‘’mL’’ and not ‘’ml’’. check that throughout the text

Response: We revised the unit of mL throughout the manuscript.

  • Provide city and country for the equipment

Response: We provided the city and country of the equipment where necessary in whole manuscript.

  • Provide full name for all the abbreviations first time mentioned in the text.

Response: As suggested we added the full name of abbreviation where that was firstly mentioned in the manuscript.

Reviewer 2 Report

In this work, authors coated urea with zinc sulphate (ZnS) and ZnS plus molasses (ZnSM) to control its N release, decrease the ammonia (NH3) volatilization and improve N utilization efficiency by sunflower. Results showed coating urea with Zn plus biopolymer is recommended to fertilizer production companies for improving N utilization efficiency, crop yield, and reducing urea N losses to the environment in addition to fulfill crop micronutrient demand.

This work is interesting and this manuscript is well prepared.

Some suggestion for the authors:

  1. "N" in the title should be written in full.
  2. L84: Introduced superabsorbent materials here. What kinds of macronutrients are provided by superabsorbent materials?
  3. “Molasses” should be introduced in the introduction part.
  4. L124: It should be “Daejung Chemical & Metals Company”.
  5. Label the main peaks in the IR of ZnS and ZnSM (Fig. 2d). Check whether the peak of 2810 cm-1 was corresponded to O-H stretching vibration. Authors should compare the IR of UC with ZnS and ZnSM.
  6. In Fig. 2e, authors should indicate the peak of coating materials.
  7. In Figs. 3-7, authors should delete the ANOVA tables.
  8. I suggest the authors combining results and discussion.
  9. The format of references should obey the rules of journal.

Author Response

Response to Reviewer#2 Report

Manuscript ID: polymers-1365278

Title: Zinc plus biopolymer coating slows N release, decreases ammonia volatilization from urea and improves sunflower productivity

Dear Editor,

On behalf of the co-authors, I appreciate and thankful for the useful comments and suggestions raised by the respected reviewers #1 and #2 on the previous version of the manuscript. Hereby, I am resubmitting the manuscript after fully taking into account the comments and questions raised and the corrections proposed by the reviewers. The corrections are presented in track changes option in the manuscript. Authors are very thankful to the reviewers and editor for their positive comments. The point-by-point answer to the comments and questions of the reviewers could be found below:

Reviewer#2

In this work, authors coated urea with zinc sulphate (ZnS) and ZnS plus molasses (ZnSM) to control its N release, decrease the ammonia (NH3) volatilization and improve N utilization efficiency by sunflower. Results showed coating urea with Zn plus biopolymer is recommended to fertilizer production companies for improving N utilization efficiency, crop yield, and reducing urea N losses to the environment in addition to fulfill crop micronutrient demand.

This work is interesting and this manuscript is well prepared.

Response: We are very thankful to the respected reviewer for the positive remarks about our study and the manuscript.

Some suggestion for the authors:

  1. "N" in the title should be written in full.

Response: Revised accordingly.

  1. L84: Introduced superabsorbent materials here. What kinds of macronutrients are provided by superabsorbent materials?

Response: Superabsorbent materials are introduced in the text now.

There was a mistake in the sentence which is now revised as “They do not only control urea release rate but also provide macronutrients such as sulfur or Zinc (Zn) to the plants if these nutrients are blended in the coating materials and thus improve soil fertility”.  Please consult lines 90-91.

  1. “Molasses” should be introduced in the introduction part.

Response: We added the introduction about molasses in the introduction section now. Please consult lines 84-89.

  1. L124: It should be “Daejung Chemical & Metals Company”.

Response: Revised accordingly. Please consult lines 132-133.

  1. Label the main peaks in the IR of ZnS and ZnSM (Fig. 2d). Check whether the peak of 2810 cm-1 was corresponded to O-H stretching vibration. Authors should compare the IR of UC with ZnS and ZnSM.

Response: We label the IR main peaks in the fig. 2d. Thank you for correcting this error. We revised the 2810 cm-1 peak description in the updated version and added the comparison of UC with ZnS and ZnSM treatments. Please consult lines 319-326.

  1. In Fig. 2e, authors should indicate the peak of coating materials.

Response: We label the peaks in the fig. 2e.

  1. In Figs. 3-7, authors should delete the ANOVA tables.

Response: We respect your opinion but removing the ANOVA table from the figures mean we are removing the statistical analysis from the manuscript which is not good. Moreover, addition of the statistics in the figure is make it more attractive and easy to understand, therefore for the sake of clarity it is good to keep the ANOVA table inserted in the figure.

  1. I suggest the authors combining results and discussion.

Response: The current structure is in accordance to the author guidelines for the journal, it can be combined with results but for the clarity we want to keep these section separate.

  1. The format of references should obey the rules of journal.

Action: The references are revised to adhere with journal style. Please consult reference section.
